# Resentful, Resigned and Respectful: Opioid Analgesics, Pain and Control, a Qualitative Study

**DOI:** 10.3390/pharmacy13010025

**Published:** 2025-02-11

**Authors:** Richard Cooper, Catriona Matheson, Emily Pagan, Helen Radford

**Affiliations:** 1School of Medicine and Population Health, University of Sheffield, Sheffield S10 2HQ, UK; emily.pagan@nhs.net (E.P.); helenmradford25@gmail.com (H.R.); 2Centre for Healthcare and Community Research, University of Stirling, Stirling FK9 4LA, UK; catriona.matheson@stir.ac.uk

**Keywords:** opioid analgesics, chronic pain, qualitative research

## Abstract

Opioid analgesic prescribing has increased significantly with associated concerns about dependence and overdose. This study aimed to explore non-cancer patients’ experiences and views of taking opioid analgesics to manage their pain. Twenty-two patients were purposively sampled from English GP practices and participated in semi-structured telephone interviews. Braun and Clarke’s thematic analysis was used to generate emerging latent and semantic themes. Patients resented taking opioid analgesics due to tolerance and addiction fears but were resigned to experiencing chronic pain. Control emerged in relation to patients’ acceptance of doctors’ control over treatment decisions but also patients’ attempted self-control over medicine adherence. This involved negatively perceived attempts to control pain but also prevent tolerance and addiction. Non-pharmacological treatments were viewed negatively by patients and addiction awareness arose from various sources. Patients were respectful of doctors but expressed negativity about the lack of addiction warnings, medication reviews and appointments. Family and friends were infrequently mentioned, as was reference to shared decision-making, suggesting patients navigate control over opioids and pain in relatively isolated ways. Patients reported generally negative experiences of opioid use for pain, which provides key insights for health professionals to enhance understanding and the management of such patients.

## 1. Introduction

Chronic pain and the particularly optimal ways to manage it represent ongoing issues in many countries. It is estimated that around 11–20% of people in Europe and the United States (US) may experience chronic non-cancer pain [1], and estimates in the UK suggest this may be even higher, with between a third and half of the population experiencing a type of chronic pain [2]. These are linked to increasingly ageing populations and associated chronic conditions. Opioid analgesics play a significant role in the management of various types of pain, ranging from traditional opiates such as morphine and codeine to more recent semi-synthetic and synthetic opioids such as oxycodone and tramadol. Opioid use in acute, operative and cancer pain is well established and clinically supported, but concerns have been increasingly raised about opioid use in chronic non-cancer pain [3]. Concerns relate to a variety of harms including dependence in particular but also increased risks of overdose and fatalities [4], respiratory depression and hyperalgesia, misuse, abuse and medication-error-related adverse events [5] and sociological concerns about stigma and shame also [6,7]. Exacerbating these concerns have been trends of increased prescribing and availability of opioids in many high-income countries and more specifically prescribing of higher strength opioids [8,9,10]. These trends are multifactorial, attributed partly to commercialised healthcare systems influenced by pharmaceutical companies and lobby groups leading to a

“[…] perception, promoted by some pharmaceutical manufacturers and clinical societies, that chronic pain in the general population was under-treated.”.[11] p. 2

A deficit in opioid alternatives has also been cited as a contributing factor, with other medications considered ineffective or having excessive side effect profiles, and all these factors have led to opioid analgesics being considered a global issue [10].

Studies have reported both prescribing doctors and affected patients expressing negative experiences surrounding the management of pain and associated outcomes. Doctors have recognised their need to develop prescribing skills, opioid pharmacological understanding and patient communication skills [12,13]. Patients using opioids have unmet needs with regards to pain relief and support in improving quality of life [12]. For other health professionals such as pharmacists, further negativity about opioids and pain management has been reported, relating to over-prescribing and limited clinical guidance [14]. Patients perceive opioids to be potent and associated with addiction and dependence but also express dissatisfaction with alternative therapies [15,16]. A recent meta-ethnography of international qualitative studies exploring patient experiences of opioid use in chronic pain identified themes such as reluctant use, good and bad understanding, problems in the therapeutic alliance, stigma, tapering and withdrawal challenges [1]. This review identified only two studies relating to the UK, where prescribing and use trends may not reflect those in other high-income countries. This study aimed to address this relative lack of understanding and sought to explore the experiences and views of patients in England taking opioids for non-cancer pain, capturing a range of demographics and clinical aspects such as patient location, age, clinical condition, opioid type and also medication dependency status.

## 2. Materials and Methods

This qualitative study was one phase of a wider study of non-cancer opioid analgesic use among patients in England, which involved an initial cluster sample of patients from 10 GP practices across England. Practices were selected to represent a broad distribution geographically but also in terms of the number of patients registered at each GP practice site (ranging from around 4600 to over 19,000 patients), urban and deprivation, reflecting recognised patterns and variations in opioid prescribing across England [9,17].

Participants in the quantitative phase of the study were invited to complete a postal questionnaire, which included a dependence measure—the Prescription Drug Use Questionnaire Patient version (PDUQp) [18]—and the option to participate in the qualitative interviews presented in this paper. From those who responded, a purposive sample of patients (based on age, sex, GP practice, dependency status and score) was invited to participate in a semi-structured telephone interview. Inclusion criteria included patients currently taking an opioid for analgesic purposes for non-cancer pain for a period of at least 3 months and having the capacity to consent. A qualitative interview guide was developed based on an initial review of the relevant literature and an analysis of quantitative stage questionnaires (see Appendix A). Topics included exploring in more depth patients’ conditions and their use of opioid medicines over time and whether patients considered them (and other treatments) to be effective in controlling their pain, their experiences of health and social care services and the impact their opioid use has had upon key aspects of their lives such as relationships, work and other activities. The telephone interviews were fully audio-recorded with patients’ consent from a private university meeting room (using an in-line digital audio-recording device). Audio recordings were then transferred to a secure university digital file store, and then trained university transcribers used them to produce an anonymised written transcript; this was used in the subsequent analysis using Braun and Clarkes’ six-stage thematic analysis [19,20] to identify relevant themes. Theoretical saturation was used to determine the final sample size with additional participants being identified and recruited until no new themes emerged from the analysis; interviews were undertaken around January 2018. Coding was undertaken manually using the annotation of paper transcripts and the charting of emerging themes, which were reviewed and revised as the analysis progressed, using the later stages of Braun and Clarke’s 6-stage thematic analysis. In particular, active attempts were made to identify not only semantic themes (those more explicit and literally identified within the data) but also latent themes (those that were less literal and reflected more underlying themes).

## 3. Results

The analysis revealed a range of themes reflecting experiences with healthcare, and doctors in particular, along with experiences of living with chronic pain and the role of various treatments. Three dominant latent themes of resentment, resignation and respectfulness emerged. Patients were resigned to taking opioid analgesics yet resented this fact while appearing implicitly respectful of the clinical decisions of doctors. Contrasting aspects of control also emerged as a further latent and overarching theme: patients relinquished control of medicine initiation and dose titration to doctors while attempting to gain subsequent control over how they consumed and adhered to their opioid medication. The latter was a balance of the perceived threat of dependence with the need to control pain; many of these were captured in one key quote, from Kim and her account of long-term opioid use for post-operative knee pain:

“[…] I think I did get myself off the pills but then I got in so much pain they put me back on again […] and I’ve always sort of not been good at taking them, I do take them when the doctor says you have to, but when you’ve been on them for a very long time you think: ‘have you been on them too long? Are they doing anything?’ And try to wean off.”Kim

Each theme is now considered in more detail with illustrative quotes from different patients. As Table 1 indicates, all participants were currently taking an opioid but several reported using other analgesics, and these are captured in the accounts that follow.

### 3.1. Resigned to Pain

Patients were resigned to experiencing pain, despite the use of opioids, additional analgesics and other therapies. Views about pain varied but most demonstrated stoicism towards pain being unavoidable and only partially treated:

“I’d like to say I’m on this painkiller and that painkiller and it’s doing the job. At the moment nothing’s touching it […] I grin and bear it and I shouldn’t. I’m not one to complain […]” Dan

Pain was not a static phenomenon and patients with chronic conditions described worsening of pain along with partially predictable fluctuations, for example, after exercise. Controlling pain was challenging with both dependent and non-dependent patients needing to be vigilant about opioid timing to avoid excess pain:

“I am in a lot of constant pain but at least this sort of takes the edge off it a bit. If I forget to change it [fentanyl patch] and I do know. I am in a lot of pain. I realise I’ve forgotten to change the patch. But I make sure I’ve got a thing set on my phone that reminds me to change it every seventy-two hours.” Katie

Patients appeared to legitimise their use of opioids by comparing their pain to prior painful experiences such as obstetric pain and migraines. By the pain being more severe than these episodes, this seemed to give their opioid use context and validation. Although some patients reported acute accidents and iatrogenic harm, most patients received a chronic medical diagnosis which initiated their illness narratives and provided further legitimation for opioid use. Also linked to resignation, there was a pessimism about prognoses and the future, often reinforced by doctors:

“Basically I’ve got a worn out disc now […] So the doctor has said it is never going to get better. It’s just something basically that I’ve got to kind like live with. So I live with back pain like every single day, but some days it is worse than others.” Mike

For a minority of patients, particularly those with more acute conditions, there was more optimism, and patients either reported that they no longer used opioids, or felt that their use might reduce over time.

### 3.2. Resentment of Medicines 

The most explicitly articulated concern for patients was a negativity towards taking any medicines but opioids in particular in the accounts of both non-dependent and dependent patients: 

“I would love to be off the medication…absolutely I hate taking them. Absolutely hate it. But I know I can’t function without them.” Kara

Patients actively resisted consumption of medicines even if it meant they experienced pain, as Len noted about his attempts to control migraines: 

“I am not a good tablet taker. I would prefer to suffer for twenty minutes if you know what I mean and then take a tablet. But if it’s…I know there’s a migraine coming on, I have to take something because I know for a fact it’s going to knock me out you know.” Len

Most patients reported active attempts to limit opioid doses and even try to stop completely, often accepting higher pain levels in order to reduce dependency risks. Patients were generally knowledgeable about their opioid, reflected by the use of generic and brand names interchangeably, dose and strength specifics and many referring to the ‘maximum’ dose they typically never exceed in their accounts.

Opioid side effects contributed to resentment, with participants demonstrating well-informed lay knowledge of common complaints. The terms ‘dependence’ and ‘addiction’ were explicitly used along with implicit concerns about the loss of therapeutic effects and fears of becoming ‘immune’:

“[…] I also keep changing them if I can. If I’m on one for quite a long time I’ll switch to a different one because I think your body gets used to it.” Vera

For several patients, the recognition of being on the maximum strength of an opioid carried anxiety due to the lack of future pain control. The concern that the prospect of addiction instilled varied. Those with experience of withdrawal symptoms or previous addiction to either alcohol, illicit substances or prescription medications had heightened concerns:

“So I only take them if I’m actually in pain and it’s really, really annoying […]. I was a very, very severe alcoholic. Obviously there’s an addiction…underlying addiction problem so I’m trying to keep off [medicines] except for my diabetes medicines and my statins.” Clive

All patients recognised that opioids have addiction potential; however, some perceived themselves to have non-addictive personalities and were not concerned whereas others, and particularly those who had other previous addictions, considered this a key concern. Opioid addiction awareness came from a range of sources, including prior knowledge, internet searches, social media, general media, friends and family and, for some, medical advice (see Table 2).

Often, multiple sources contributed to patients’ lay understanding of opioids and addiction, as Clive summarised: 

“The information I’ve got is via the internet, via the newspapers but I don’t tend to believe what the newspapers print. I’d rather double check with the NHS. I do know people do get addicted and are compulsive pill poppers so I’ve seen that in my dad [and my], girlfriend’s mum she used to be a compulsive pill popper” Clive

Family and partners emerged in several patient accounts; some served as cautionary examples as in Clive’s case or had commented negatively and had questioned patients’ use of opioids as was the case for some friends and work colleagues: 

“My partner doesn’t like me taking things like that. She’s very against all forms of medication really, generally, especially painkillers. But that’s just her, that’s just her opinion.” John

Despite patients’ relatively informed knowledge of opioid strengths and dosages, many did not have concomitant pharmacological insights. Several patients noted that it was only from hearing about their medicines in the media that they had become aware they were taking opioids: 

“Well I knew opioids were addictive but I just didn’t realise until it was on the news. I didn’t realise that the fentanyl was an opioid […] I know what opioids are like cocaine and stuff like that.” Katie

Several patients had actively attempted to either reduce the dosage or completely cease opioids with some reporting withdrawal symptoms which, particularly when linked to embodied physical effects, heightened their addiction concerns. Withdrawal symptoms varied and included flu-like symptoms, headaches, restlessness, shaking, insomnia and nausea. In some cases, doctors appeared to confirm and legitimise such experiences: 

“[…] when I was poorly in the hospital and sort of fidgety and twitching and this particular doctor said well it’s obvious because you’ve stopped codeine for four days.” Laura

Drowsiness was most often mentioned, however, and several patients reported this being an increasing problem with stronger opioids and tramadol in particular. Several patients also reported the associated impact this had on their work and productivity: 

“I never took tramadol while I was at work. I just tried to manage on the codeine because obviously tramadol makes me very drowsy […] so yeah it was just trying to balance what kept the edge off the pain enough to be able to do my job really.” Claire

Tramadol was viewed particularly negatively, and several patients recounted how it had been discontinued by prescribers due to adverse side effects. Despite the frequency of musculoskeletal conditions, there was a surprising lack of reference to NSAIDs and paracetamol. Some patients reported non-opioid analgesics helped control pain and other symptoms like insomnia; however, they were often regarded as being ineffective:

“I…the GP decided that I’d been on things quite long enough, and they changed them all so I has something else…yes I mean I was on paracetamol and you might just as well throw them in the bin, […] they have got no effect whatsoever.” Sylvia

This quote also illustrated the recurrent description of patients appearing to passively accept medical decisions in the correction of ‘I’ to ‘the GP’, which will be considered later in terms of ‘respect’ for doctors.

### 3.3. Other Treatment Options 

A further factor that appeared to substantiate the previous two themes of resignation to pain and resentment to taking medicines was the perceived lack of effective opioid alternatives. Patients identified several non-opioid treatment options; some viewed these positively, yet overall, they were viewed negatively (Table 3). Some patients attributed this to a lack of perceived benefit while others reported logistical issues such as referral/appointment delays which caused some to give up trying to access such services. Sharon described her request for alternative analgesia; however, as will later be described, doctors retained control over this decision, and Sharon appeared to accept their input:

“I asked my new GP. I said ‘Could I get something stronger?’ She said ‘I am so sorry Sharon’ she went, ‘but we can’t. We can’t give you something stronger’; [she] talked about physiotherapy [and] I did one session and it were absolutely brilliant. It eased the pain for a couple of weeks but then I was back to the same pain again. So really it were just to be honest it were a waste of time doing that really.” Sharon

There appeared to be variation in which patients were referred to a pain clinic. Some patients reported being reviewed by the clinic as a positive experience; however, others reported this service not being made available to them.

Patients expressed being ‘stuck’ on opioids with alternatives failing to manage their pain. This lack of pain control led to a minority of patients increasing their opioid quantities as was the case with Georgia and Veronica. Alice illustrates the concerns surrounding addiction and lack of alternatives:

“Well you know, my only concern is that I’m addicted to it and I know I will be after this length of time, but what is the alternative? […]. All the alternatives I’ve had have never done anything at all, so at least this keeps my pain level just to a stage.”Alice

### 3.4. Respectful of Doctors

Doctors were referred to repeatedly throughout patient’s accounts, the majority being general practitioners with others including hospital consultants and those involved in pain clinics and acute admissions. A key finding that emerged was implicit descriptions of patients accepting both opioid and non-opioid medical treatment passively and appearing respectful of doctors overall. However, there was criticism, with issues regarding difficulty obtaining appointments, a lack of continuity of care and medication reviews, conflicting information being given by different doctors and a perceived lack of warning about opioids and addiction overall. Overall, patients were implicitly respectful toward doctors and particularly their decisions about medicines. This was shown by their acceptance of medical paternalism and the use of the pronominal ‘they’ to generalise doctors, as Kara illustrated in her account of having medicines changed:

“I was on tramadol and then they put me on to, what was it, pregabalin, yes because the tramadol just didn’t seem to be touching and then it was the pregabalin and that that helped, […] but then I don’t know if I got immune to that as well […] and then eventually this other doctor changed pretty much all of them so I’m just coping with those.”Kara

This account contained elements that were typical of many patients, including lay references to tolerance—being ‘immune’—and reflecting a passive acceptance of doctors’ authority. Patients’ accounts did vary and ranged from some accepting doctors’ decisions, even when expressed as recommendations, to challenging and criticising them. However, across all patients, there was still an overarching acceptance of doctors’ decisions. Sylvia recounted instances where doctors made errors but countered them with repeated examples of her compliance and trust in doctors, as these two contrasting quotes illustrate: 

“[…] under no circumstances was I to have gabapentin whether it was to do with medication I was on already, I don’t know, but anyway the stupid doctor you know gave me gabapentin and I tell you my ankles swelled […]”Sylvia

“[…] you go to the GP or you are in hospital, you come out the doctor writes a prescription out for you, so you automatically think that it is safe […] and you think they know what they are doing when the give you a prescription […]”Sylvia

As the above illustrates, examples of perceived poor medical practice were reported, and accounts of iatrogenic harm arose, for example, relating to post-operative pain. Infrequent examples of shared decision making were reported and often related to GPs giving a choice of medications, as Flora described:

“Actually he gave me the choice. He said ‘You can have, I can give you, a tramadol or I can give you’—I can’t remember what the other tablet was […] I said ‘I’ve heard of tramadol a friend of mine takes it and she finds it very good so I’ll try the tramadol’ […]. That was how I actually first came to take it and nobody’s reviewed it with me since […].”Flora

This quote illustrates the influence of others in relation to opioid decision making, but also about the subsequent lack of medication review. Veronica felt that this lack contributed to her escalating use of codeine at doses significantly higher than recommended:

“No, I feel angry in a way because they could have stopped it a long time ago. And I think if they’d reviewed me more regularly they could have probably picked up before even I did, but there was an issue. I mean because by the time I’d picked it up, I was going to go into withdrawal. And then I had no support when I was going through withdrawal either.” Veronica

For other patients, reviews were reported and appeared to involve relevant discussions about opioids, but this was less common and in examples such as Mike’s below, may have been related to his frequent contact with his GP: 

“[…] when I went in for a review […] she said you are not a red flag alert to us really because she said there are some people she said that like take thirty to forty Solpadeine a day with an addiction. And I was actually in shock. I was just like wow. I said I’ve never gone past eight a day.”Mike

Many patients expressed negativity about the lack of warning about possible addiction given, and at times, conflicting medical advice. For some, this led to anger about not being able to make an informed decision about their medicine:

“I felt like the doctor should have said you maybe come addicted or what the problems could have been and then I might have said ‘No I’m not taking them. I’ll take an alternative’. […] nobody seems to tell you things these days about… not just tablets, but you have to find a lot of information out yourself.”Kara

Other healthcare professionals were rarely mentioned, for example there was brief mention of community pharmacists; however, pharmacists did not appear to represent a significant professional group, except as being the route to opioid supply.

## 4. Discussion

Overall, patient experiences of opioids varied greatly. There were overriding reports of resignation to being in pain alongside strong resentment to requiring opioid medications to only partly relieve pain. Patients appeared to respect and accept the decisions of doctors to initiate or change their treatment, including opioid-based and alternative treatments, while maintaining a broadly negative opinion as to whether their pain could be ameliorated. Unfortunately, the support available to patients requiring analgesia varied widely based on these patient narratives. This included inconsistencies regarding who was referred to pain clinics, the extent of information that was provided to patients about what medicines they were prescribed and warnings about addiction and tolerance risks. In response to concerns about tolerance and addiction, patients attempted to exert control over their own use of opioids, often reporting trying to take them less frequently or only when in pain. This highlights the complexity of the relationship between patients and their opioid analgesia and the various ways they relinquish and retain control. Figure 1 illustrates key themes and where they relate to the respective patient or doctor domains and overlaps. As the figure shows, there were few examples of a genuine relationships and joint decision making between doctors and patients.

## 5. Comparison with Existing Research

This study highlighted the largely negative experiences and attitudes patients had towards pain management, often describing perceived futility regarding attempts to control pain. This attitude has been documented in prior qualitative studies and meta-ethnographies also describing patients returning to medical professionals with ongoing pain despite increasing doses of analgesics [1,12].

A proportion of patients enrolled in this study emphasised the lack of information regarding opioids they were given prior to taking them. Although some patients demonstrated a high level of opioid-based knowledge, some highlighted the perceived lack of warnings regarding addiction and tolerance they received which may have altered their decision to commence opioids. This lack of patient awareness regarding opioids and their risk is seen in other literature, also extending to a lack of awareness of support available for those suffering from dependence [21]. The need for patient education is also implied by efforts, both in the UK and internationally, to educate the general public about potential opioid candidates [10].

Concerningly, patients in this study also reported a perceived lack of support after commencement on an opioid, for example, a lack of medication review appointments. This finding was echoed in other studies reporting patients remaining on opioid prescriptions without sufficient follow-up or clear treatment plans, in some cases resulting in patients using opioids for longer than necessary, increasing addiction risk [21].

A major barrier to opioid use reduction was reported to be a lack of suitable analgesic alternatives, and within this study, this added to patient anxiety due to concerns regarding what could alleviate their pain once high-dose opioids became ineffective due to tolerance. This lack of suitable alternatives has been reported in other studies that also reference it as a catalyst for long-term opioid prescribing [1,10,12].

### 5.1. The Role of Doctors in Opioid Analgesic Dependence

A key emerging concern from patients in this study was that whilst they appeared to be respectful of doctors in terms of accepting their prescribing decisions, negativity did emerge in relation to several key issues, including a perceived lack of sufficient addiction or dependence warning given to patients, a lack of review or monitoring of their opioid prescribing and difficulty obtaining appointments and continuity of care. In relation to providing warning, this was arguably related to the other issue, summarised in Figure 1, of the doctor–patient relationship and the lack of emerging examples where this appeared to involve shared decision making and communication. Similar themes and concerns have emerged in previous qualitative research involving opioid analgesic patients and GPs in England [12] and a meta-ethnography of pain patients [22]. For McCrorie et al. [12], a concern about ‘locating control’ and differing doctor and patient perspectives was identified. Evidence does appear to suggest opioid medication reviews are undertaken, and Song and Foell [23], for example, reported an audit of opioid analgesic prescribing, and reviews were documented in 85.7% of cases, but the quality of such reviews could not be assessed. RCGP guidance materials [24] also suggest the need for appropriate monitoring as part of the prevention of misuse and dependence and specifically describe what should be involved in ‘discussions with patients’, and in particular, ensuring patients are given sufficient information and warning about dependence.

### 5.2. The Role of Pharmacies in Opioid Analgesic Dependence

There was surprisingly little mention of community pharmacies by participants in this study. Indeed, the role of pharmacists and pharmacy staff seemed to be considered as one of supply with little clinical input; although, there was one passing reference to a pharmacist reviewing a patient’s medication. This differs from the wider policy and research context in which the role of both community and primary care (GP practice) pharmacists in managing chronic pain patient medication has been researched, and evidence suggests there could be clinical benefit from pharmacist involvement. Bennet et al. [25] undertook a systematic review of pharmacist-delivered educational interventions in chronic pain management, which included four studies in a meta-analysis. The findings demonstrated a reduction in average pain intensity (0.5 on a 0 to 10 scale), a reduction in adverse effects by more than 50% and an improvement in satisfaction with treatment (1 point on a 0–10 point scale). Community pharmacies would appear to be an obvious location to deliver an educational intervention, given that people attend regularly to collect prescriptions, and there is a documented lack of treatment satisfaction in the current model of care, which was noted in this study and elsewhere. A Canadian study [26] found patient satisfaction with pain treatment was low, particularly around the provision of information regarding treatment and medication. That study concluded community pharmacists could extend their role to improve the management of chronic non-cancer pain. In a GP practice setting, an exploratory trial by Bruhn et al. [27] indicated that pharmacist medication review (with or without pharmacist prescribing) could reduce pain intensity and improve mental wellbeing in patients with chronic pain. Several evidence reviews have also identified opportunities for pharmacists to contribute to opioid stewardship, leading to beneficial outcomes in areas such as education and medication therapy adjustments [28,29]. In England, there are community pharmacy services such as the New Medicines Service (NMS), which allows pharmacists to undertake reviews on certain medicines when they are initially prescribed, but none of the current eligible conditions for NMS would cover opioid analgesics. There was also a Medicines Use Review (MUR) service, which had been argued to be of relevance to managing opioids [30], but this service was discontinued in England in 2021. It should also be noted that pharmacists are increasingly undertaking prescribing in several countries, and this research hopefully provides insights for pharmacists who may be involved in the prescribing of opioids and the management of pain.

## 6. Strengths and Limitations

This study had key strengths in linking patients’ self-reported quantitative opioid use and providing additional insights linked to dependency status and experience, with no obvious patterning of experiences or views linked to whether patients were dependent or not according the PDUQp definition. Purposive sampling captured a range of different demographic characteristics across a number of GP practices in England. Interviews were by telephone, and this may have impacted the establishment of rapport in the interviews but were preferred by participants. The interviews were conducted around January 2018 and reflected the prescribing trends and service provisions in England at that time and may not reflect current practices. However, they remain a powerful and important insight into patients with chronic pain taking opioid medicines.

## 7. Conclusions

This paper reveals that patients have complex relationships with opioids. This study offers further evidence of problematic opioid use and of patients resigned to pain, resenting opioid medicine consumption, but being respectful of doctors and managing in relatively isolated ways. Different aspects of control also emerged, which were located in medical authority but also patient autonomy with a contested overall balance in relation to this. There are several implications for clinical practice and policy in relation to the need to increase awareness of opioid addiction risks among the public, as numerous other studies have found, to improve the appropriate prescribing and also deprescribing, improve the reviews on opioids as well as associated reviews on the management of non-cancer chronic pain more generally and increase awareness on how shared decision-making can be achieved between patients and various health professionals. This in turn suggests important opportunities for other health care professionals to do more and to review their relationships and communications with patients on opioid analgesics to improve their experiences.

## Figures and Tables

**Figure 1 pharmacy-13-00025-f001:**
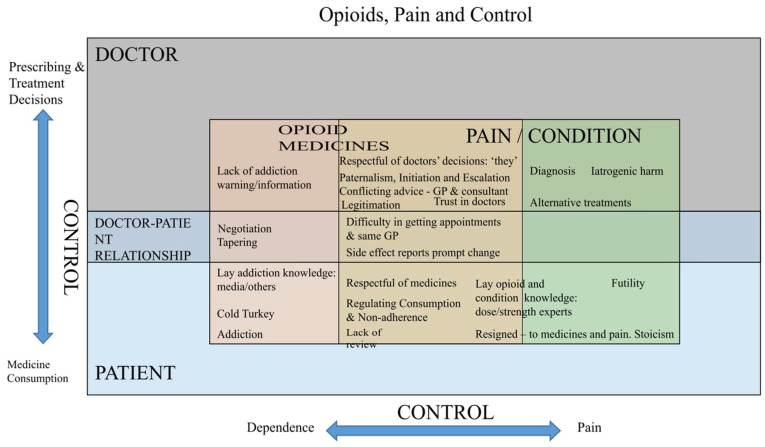
Summary of the relationship between patients and doctors.

**Table 1 pharmacy-13-00025-t001:** Summary of participant patient characteristics and relevant clinical details.

Pseudonym	Location	Age	Employment	Current Opioid and Other Medicine	Initially Prescribed by	Pain-Related Condition	PDUQp
Laura	U2C1	60s	Retired	Codeine, co-dydramol	Hospital gastroenterologist	Pain and associated symptoms of Irritable bowel disease	16
Louise	U2C1	60s	Retired	Codeine, gabapentin, amitriptyline, co-codamol	Hospital pain clinic	Osteoarthritis, knee replacement, post-op complications	16
Georgia	U2C1	50s	Retired	Oxycodone, codeine	GP	Back pain	15
Elizabeth	U2C1	40s	Long term sick/disabled	Co-codamol	Nurse prescriber	Renal calculi	14
Mike	U2C1	40s	Self-employed part-time	Co-codamol	GP	Accident at work, then car accident	13
Sylvia	U1A1	70s	Retired	Buprenorphine dihyrocodeine parcetamol	GP and hospital	Accident/pain injury, polymyalgia rheumatica	13
Tony	U1A1	50s	Long term sick/disabled	Buprenorphine	Not disclosed	Accident	12
Dan	U2C1	60s	Retired	Zomorph, co-codamol, co-dydramol	GP and hospital	Accident/fractures	12
Kara		60s	Employed	Tramadol	GP	Back pain	11
Vera	U2C1	80s	Retired	Co-codamol	GP	‘Severe arthritis’	11
Kim	U2C1	60s	Long term sick/disabled	Tramadol, co-codamol, gabapentin, amitryptline,	GP	Arthritis, knee operation	10
Clive	U1A1	50s	Long term sick/disabled	Co-codamol	GP	Arthritis ‘joint pain-hips, knee, ankles, shoulder’	10
Claire	U1A1	50s	Long term sick/disabled	Co-codamol	Hospital surgeon	Posterior tibial dysfunction	10
Sharon	U1B1	20s	Long term sick/disabled	Co-codamol	GP	Sciatica—Herniated Disk	8
Alice	U1A1	70s	Retired	Morphine	GP	Degenerative disc disease, osteoarthritis/knees, arachnoiditis	7
Katie	U2C1	60s	Long term sick/disabled	Fentanyl	GP	Degenerative lumbar disc disease	6
Jackie	U1B1	70s	Retired	Co-dydamol	GP	Arthritis spine, spondylosis neck	6
John	U1A1	60s	Retired	Co-codamol, Tramadol	GP	Osteoarthritis, rheumatoid arthritis	3
Veronica	U2C1	30s	Employed	Co-codamol	GP	Sciatica after hysterectomy operation	3
Jack	U1A1	60s	Self Employed	Codeine	GP	Back pain	2
Len	U1B1	70s	Retired	Co-codamol	GP	Migraine	1
Flora	U1A1	60s	Retired	Tramadol	Hospital registrar	Rotator cuff injury	1

U1A1: large urban area, major conurbation; U1B1: large urban area, minor conurbation; U2C1: smaller urban area, urban city and town.

**Table 2 pharmacy-13-00025-t002:** Influences of the social construction of opioids and addiction.

Negative Depiction	Positive Depiction
Medical advice	Medical advice
Personal experience	Personal experience
Media reporting	
Social media	
Internet	
Celebrity addiction	
Experiences of family and friends	

**Table 3 pharmacy-13-00025-t003:** Alternative treatment options.

Treatment	Positive Aspects	Negative Aspects
Physiotherapy	Few but some short-term benefits	Made symptoms worse, long waiting list, lack of any benefit, low motivation to continue
Acupuncture	Fear	None reported
Non-opioid analgesics	Non-addictive	Tolerance, concerns, side effects and contra-indicated
Pain clinic	Improved pain	Waiting lists
Self-management	Prevented exacerbations of pain, linked to maintaining mobility	Cost (of equipment), required motivation
Psychological therapies	None reported	Patronizing and not effective

## Data Availability

The data presented in this study are available on request from the corresponding author due to the nature of the qualitative interview data collected. Transcripts were fully anonymised but still contain personal details that make them unsuitable for general availability.

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
