# Peer review of "Resentful, Resigned and Respectful: Opioid Analgesics, Pain and Control, a Qualitative Study"

_pharmacy, 2025, doi:10.3390/pharmacy13010025_

Round 1

Reviewer 1 Report

Comments and Suggestions for Authors

The manuscript submitted for review concerns the subjective feelings of non-oncological patients during pharmacotherapy of pain.

Comments on the research methodology. It seems advisable to present the methodology for selecting respondents for the telephone interview – it is not clear to the reader how this selection was made based on the presented criteria, such as gender, age, medications used, the actual condition of the patient. Statistical data should be presented, i.e. what proportion of respondents had access only to a family doctor, how many used consultations in pain treatment departments in hospitals, from which centers did they come – rural, provincial, metropolitan – agglomeration? Please discuss or attach in additional materials the protocol of questions asked to respondents during the telephone interview, the reader learns the answers to questions that were not asked in the content, so it is difficult to refer to the accuracy of the conclusions and theses. After reading the patients' statements, a very sad picture of a sick person comes to mind, stigmatized by the disease and opioid therapy, lonely, without psychological support, left to live in suffering and fear of uncontrolled pain. A patient who respects doctors, submits to therapeutic decisions without faith in their effectiveness is a sad picture of our times. The interviews lack information/questions about the possibility of removing the cause of the pain that accompanies the respondents. Did the interviewers ask such questions? Is the patient undergoing treatment? The reader does not know. It is not known whether there is a chance for the respondent's health to improve - have the treatment options been exhausted and the only thing left is to take a painkiller? Summary. The conclusions drawn by the authors regarding the lack of doctor-patient dialogue, the lack of effective psychological support, the lack of information about the risks of long-term opioid therapy do not lead to a final conclusion. What is the authors' view on changing the described situation regarding cooperation in the field of drug reviews in the doctor-pharmacist-patient relationship. The reader does not know whether UK pharmaceutical law, being outside the EU structures, allows for contracting pharmaceutical services in community pharmacies, allowing pharmacists to perform drug reviews in non-hospital conditions. The subject of the manuscript is very important in the context of an ageing society, but the manuscript requires improvement, supplementation of the criteria for selecting respondents, approximation of the topics of conversation with interviewers and statistical data.

Author Response

Reviewer comments

Responses/changes made in revised manuscript

Thank you reviewer 1 for all your very helpful and insightful comments and also suggestions. We have responded to each substantive point below and also made changes to the revised paper (highlighted in yellow) to show where these have been made but also in the comments below.

1) Comments on the research methodology. It seems advisable to present the methodology for selecting respondents for the telephone interview – it is not clear to the reader how this selection was made based on the presented criteria, such as gender, age, medications used, the actual condition of the patient. 

Apologies for not stating this in more detail but this qualitative stage of the wider project drew on participants providing contact details when they responded with their completed survey. The main survey asked for key demographic details including: age, gender, ethnicity, and we used post code and highest educational level as proxies for socio-economic status. We did not cross-reference medical records as this would have required another level of ethical approval in the UK to access this level of personal data; the only main inclusion criteria that GP practices were asked to use were as follows: must have been on an opioid medicine for at least 3 months at the point the GP practice records were searched, that the opioid be used for pain (cf diarrhoea or cough in the case of codeine occasionally), that the patient not have a cancer diagnosis (ie the student related to the use of opioids to treat non-cancer pain, have capacity to consent. When survey respondents also replied to offer to participate in interviews,  we compiled an initial purposive sample frame, to capture a range of different ages, both male and female, self-reported opioid type (we wanted to capture weak, moderate and potent opioids), self-reported condition relating to opioid use).

2) Statistical data should be presented, i.e. what proportion of respondents had access only to a family doctor, how many used consultations in pain treatment departments in hospitals, from which centers did they come – rural, provincial, metropolitan – agglomeration?

We have provided additional information on the demographics of respondents. We were mindful based on current guidance on the ethical and anonymous reporting of participant details not to use too many identifying characteristics. We recruited from 10 GP practices in this stage of the research and the practices were classed as: large urban area (U1), major conurbation (A1) (n=2), large urban area U1, minor conurbation (B1) (n=1), smaller urban area (U2), urban city and town (C1) (n=6). A final practice was designated Accessible settlement R1, rural town and fringe D1 but we did not receive any requests to participate from this practice.  Based on the study design and sampling via GP practices, all the patients were recruited by virtue of being registered with a family doctor (GP); some reported having additional care from hospital/secondary care services (predominantly pain clinics but some related to clinical manifestations) and also other primary/community services such as physiotherapy.

We have also included self-reported details of who initially prescribed the opioid analgesic for each patient where  this was known from the triangulated survey data where it was one of the questions. This also arose in many interviews.

3) a) Please discuss or attach in additional materials the protocol of questions asked to respondents during the telephone interview, the reader learns the answers to questions that were not asked in the content, so it is difficult to refer to the accuracy of the conclusions and theses. 

b) After reading the patients' statements, a very sad picture of a sick person comes to mind, stigmatized by the disease and opioid therapy, lonely, without psychological support, left to live in suffering and fear of uncontrolled pain. A patient who respects doctors, submits to therapeutic decisions without faith in their effectiveness is a sad picture of our times. The interviews lack information/questions about the possibility of removing the cause of the pain that accompanies the respondents. Did the interviewers ask such questions? 

c) Is the patient undergoing treatment? The reader does not know. It is not known whether there is a chance for the respondent's health to improve - have the treatment options been exhausted and the only thing left is to take a painkiller?

a) We have provided the initial interview schedule which was approved by the NHS Research Ethics Committee. 

b) There was a specific question and follow-up relating to the participant’s view about the management of their pain: To what extend do you feel your current opiate medication is appropriate for your condition and symptoms. Prompt if necessary about: a)     whether the opioid (or other treatments) have managed to control the pain; b)      if there have been any side effects. We have added more to this in the methods also: ‘and whether patients considered them (and other treatments) to be effective in controlling their pain’

c) Apologies if this was not clear but as per point 2 above, an inclusion criteria was that patients were currently taking an opioid for pain relief and when prospective participants were approached to participate by an initial phone call, this was confirmed ie ‘undergoing treatment.’ During the actual interviews, patients were encouraged to discuss what previous and current treatment options they had experienced and to comment on their perceived effectiveness in managing pain.

We have provided the initial interview schedule which was approved by the NHS research ethics committee and also fully below:

Examples of Qualitative Interview Questions

  1. Please could you summarise your use of current opiate medication in terms of the medicines you take and for what conditions and symptoms.
    1. Prompt in relation to medical condition/symptoms
    2. Whether primary or secondary care
  2. Who has prescribed these for you?
    1. Ask for further details about continuity of care, type of prescriber etc.
  3. Please briefly indicate what previous medicines have you taken (for pain relief).
  4. To what extend do you feel your current opiate medication is appropriate for your condition and symptoms. Prompt if necessary about:
    1. whether the opioid (or other treatments) have managed to control the pain
    2. if there have been any side effects
  5. Do you have any concerns about the opiate medication you are currently taking?
    1. If so, what are these and please describe them.
    2. Follow-up to 5) and 5a): if yes: have you raised any of these concerns with anyone and if so, what happened?
  6. Do you feel you had sufficient information about your current opiate medication?
    1. Please describe why in more detail.
  7. Are you aware of the potential for some individuals to become dependent or addicted to opiate medication?
    1. If yes, please can you indicate what you understand by this and how you became aware of this.
  8. Do you feel you ever have been or currently are addicted to an opiate pain medication?
    1. If so, please could you describe why you think this and your experience.
    2. Follow-up to 8a): if yes, what help or treatment was sought and was this beneficial?
  9. Have you ever felt any anger or mistrust towards a current or previous prescriber/doctor?
    1. If so, please give details of why you felt this and how this arose and if it was resolved.
  10. Have you obtained pain medication from other sources apart from prescription?
    1. If so, please say where and why this source was used [prompt, from pharmacy purchase, internet, another person's medicines].

4) Summary. The conclusions drawn by the authors regarding the lack of doctor-patient dialogue, the lack of effective psychological support, the lack of information about the risks of long-term opioid therapy do not lead to a final conclusion. What is the authors' view on changing the described situation regarding cooperation in the field of drug reviews in the doctor-pharmacist-patient relationship

We agree with reviewer 1 on their summary of the findings and we had tried to indicate in the section on ‘the role of pharmacy in opioid analgesic dependence’ in the discussion to highlight whether interventions and other research have been undertaken. We had hoped that these would provide some examples in different countries of where future opportunities and interventions could be made linked specifically to community pharmacy. We have added a final sentence to this section to capture one of the most recent advances in pharmacy - namely prescribing - and to note that these findings may be of relevant to pharmacy prescribers also.

5) The reader does not know whether UK pharmaceutical law, being outside the EU structures, allows for contracting pharmaceutical services in community pharmacies, allowing pharmacists to perform drug reviews in non-hospital conditions. The subject of the manuscript is very important in the context of an ageing society, but the manuscript requires improvement, supplementation of the criteria for selecting respondents, approximation of the topics of conversation with interviewers and statistical data

Thank you for raising this further suggestion about the contracting of pharmaceutical services. In the UK, there is a current system for reviewing newly prescribed medicines (New Medicnes Service) within community pharmacies but to our knowledge, no specific services linked to the review of opioid analgesics.  In England, the commissioning of core and additional services in the community and primary care setting is controlled by Integrated Care Boards, and in theory, these can commission a range of services which could include drug reviews but we are not aware of any such schemes. We have added more to the discussion section relating to pharmacists about this and two systematic reviews. We have also offered a further critical reflection on the situation in England, wherein current New Medicine Service (NMS) review processes do not cover conditions linked to the use of opioid analgesics and we also made reference to the demise of potentially relevant services like Medicines Use Reviews (MURs) as opportunities to review chronic pain and opioids (Youssef S (2010) Better MURs for Patients with Chronic Pain. The Pharmaceutical Journal 284: 587-589.)

Reviewer 2 Report

Comments and Suggestions for Authors

I read with interest the paper titled "Resentful, Resigned and Respectful: Opioid Analgesics, Pain and Control, a Qualitative Study"

I have minor comments to authors, that could enhance the manuscript:

1. Line 45-46 - Despite of US is being one of the first countries to deal with opioid epidemic, pleanty of other countries, including UK have reported problems with opioids in the past. In this review, at least 12 studies identified problems (misuse, abuse) in Europe, with 2 identifying these problems in UK - https://www.mdpi.com/1424-8247/17/8/1009

Authors could explore the idea that this refers to a worldwide problem. 

2. "Sex" should be used instead of "gender" for academic and scientific purposes. 

3. Line 75 - please clarify what is the practice size. This could be something common in UK, but for overall public the idea could be clarified. 

4. How do you record the interviews by telephone? How were them transcribed verbatim? 

5. Interviews were conducted in January 2018. The data is not novel, and this is a major limitation for the paper presented. Why the gap of 7 years between the interviews and the tentative for publishing this paper? 

6. Conclusion could be a separate section, and focus more on the findings of the paper. 

Author Response

Reviewer comments

Responses/changes made in revised manuscript

Thanks very much for the helpful and insightful comments from reviewer 2; we have responded to each point individually below and also provided additional context and clarification which we hope is of help. We have also summarised where changes have been made to the revised manuscript and highlighted these in yellow in the revised submission.

I have minor comments to authors, that could enhance the manuscript:

1. Line 45-46 - Despite of US is being one of the first countries to deal with opioid epidemic, pleanty of other countries, including UK have reported problems with opioids in the past. In this review, at least 12 studies identified problems (misuse, abuse) in Europe, with 2 identifying these problems in UK - https://www.mdpi.com/1424-8247/17/8/1009

Authors could explore the idea that this refers to a worldwide problem. 

Thank you for this helpful comment and for the suggested paper and also for the need to not over-emphasise the US. To facilitate this we removed the sentence which referred very specifically to the US and the ‘opioid epidemic’ there; we had initially felt this might resonate with readers but agee with reviewer 2 that this could unintentionally highlight the US to the detriment of concerns in many other countries. We have added the additional paper to the reference list. We have added ‘and all these factors have led to opioid analgesics being considered a global issue’ and retained the key OECD report from 2019.

2. "Sex" should be used instead of "gender" for academic and scientific purposes. 

Thank you - this has now been changed in the main table to ‘sex’ on line 90.

3. Line 75 - please clarify what is the practice size. This could be something common in UK, but for overall public the idea could be clarified. 

Thank you for asking for clarification about this - in the UK, family doctors typically work in a single location - a General Practice - and this would have a certain number of registered patients linked to that practice. This is what is referred to as the practice size. The smallest in this study had around 4600 patients registered and the largest had over 19000 patients; the average in England at the time of the study was around 7500 patients. This would usually be spread across a number of doctors, but also nurses and healthcare assistants. We did report in the study that some patients expressed concerns about a lack of continuity of care and this may be related to trends towards larger practices with more patients and also a number of different GPs and other health professionals.

4. How do you record the interviews by telephone? How were them transcribed verbatim?

Thank you for asking for further details of this process, which we omitted in the first draft for brevity. We have now included further details as follows: ‘The telephone interviews were fully audio-recorded with patients’ consent from a private university meeting room (using an in-line digital audio-recording device). Audio recordings were then transferred to a secure university digital file store, and then trained university transcribers used to produce an anonymised written transcript; this was use in subsequent analysis.’

5. Interviews were conducted in January 2018. The data is not novel, and this is a major limitation for the paper presented. Why the gap of 7 years between the interviews and the tentative for publishing this paper? 

Thank you for checking on this and we did note this as a limitation in the discussion: “The interviews were conducted around January 2018 and reflect the prescribing trends and service provision in England at that time and may not reflect current practices. However, they remain a powerful and important insight into patients with chronic pain taking opioid medicines.”

To add further context, there were four key reasons the work was not formally submitted for peer reviewed journal publication sooner: firstly, other aspects of the project experienced unexpected delays and there was a significant period (around 14 months) when additional data collection and analysis was needed, which impacted on the author’s time and capacity; secondly, there was an additional period of time when a report for the funder, Indivior, was being prepared and this also took a significant period of time and it was not considered appropriate to publish until this had been completed for the funder (although they remained fully independent throughout); thirdly, the COVID pandemic did - like many research projects - also impact on the capacity and time of the researchers to write-up the qualitative study; finally, as noted in the discussion, we did feel that as qualitative data, we did not directly claim to generalise from the data and analysis, and as such even though this represents experiences and views from several years ago, it is consistent with qualitative methodologies to locate these findings in this time and the discussion hopefully makes this clear for readers.

6. Conclusion could be a separate section, and focus more on the findings of the paper. 

We have added a separate ‘Conclusion’ heading and also made a number of additional changes to this final section to ensure it reflects the findings but also communicates clearly the clear messages for policy and practice.

Round 2

Reviewer 1 Report

Comments and Suggestions for Authors

I recommend publishing the manuscript in its current form. The authors have broadly and comprehensively explained the doubts and supplemented the research data and the summary. Thank you very much for the interesting discussion on the topic of caring for a patient in suffering. The article presented for review takes up a very important social topic of universal significance.

Reviewer 2 Report

Comments and Suggestions for Authors

The authors adressed all the requested changes. No further suggestions. Accept in the current form.